# 'What does that mean?': a qualitative exploration of the primary and secondary clinical care experiences of young people with continence problems in the UK

Katie Whale,[1] Helen Cramer,[2] Anne Wright,[3] Caroline Sanders,[4] Carol Joinson[1]

[1]Centre for Child and Adolescent Health, School of Social and Community Medicine, University of Bristol, Bristol, UK
[2]Centre for Academic Primary Care, School of Social and Community Medicine, University of Bristol, Bristol, UK
[3]Evelina London Children's Hospital, Guy's and St Thomas' NHS Foundation Trust, London, UK
[4]University of Northern British Columbia, Prince George, Canada

**Correspondence to**
Dr Katie Whale;
katie.whale@bristol.ac.uk

## ABSTRACT

**Objectives** To explore the clinical care experiences of young people with continence problems.

**Design** In-depth semistructured qualitative interviews were conducted by Skype and telephone, with the addition of art-based participatory research techniques. Transcripts were analysed using inductive thematic analysis.

**Setting** Primary and secondary care in the UK.

**Participants** We interviewed 20 participants (9 females, 11 males) aged 11–20 years. There were six participants with bedwetting alone, five with daytime wetting alone, five with combined (day and night) wetting and four with soiling.

**Results** We identified four themes: appointment experiences, treatment experiences, engagement with treatment and internalisation and externalisation of the continence problem. Patient-focused appointments using age-appropriate language were highly desirable. Continuity of care was highlighted as an important aspect of positive clinical experiences; however, this was found to be rare with many participants seeing a different person on each visit. Participants had tried a wide range of treatments for their continence problems with varying degrees of success. Relapse and treatment failure were common. Experiencing relapse was distressing and diminished participants' belief in the success of future treatments and undermined adherence. Participants would be seen to adopt two opposing coping strategies for dealing with their continence problem— internalisation and externalisation.

**Conclusion** Incontinence in young people is challenging to manage. Young people may need to try a range of treatments before their symptoms improve. Due to challenges in treatment, there is an increased risk of poor adherence. During patient-focused appointments, clinicians should work to build rapport with patients and use age-appropriate language. Involving young people in their own care decisions is important. The way in which young people understand their continence problem can influence their coping strategies and adherence to treatment regimes.

### Strengths and limitations of this study

▶ In-depth qualitative interviews provide a unique insight into the experiences of young people with incontinence.
▶ Due to the sensitive nature of this topic, interviews were combined with an arts-based participatory approach to facilitate greater disclosure and gain a better insight into young people's lives.
▶ Skype and telephone interviews allowed access to participants from a wide geographical area.
▶ Telephone interviews presented challenges in building rapport and body language with the participants, which may have hindered some personal disclosure.

## INTRODUCTION

Continence problems (daytime wetting, bedwetting and soiling)[1] are among the most common chronic conditions of childhood. In the UK, it is estimated that 900 000 children and young people experience some form of continence problem.[2] This is, however, likely to be an underestimate as the stigma associated with incontinence often prevents individuals from seeking treatment.[3 4] Epidemiological studies estimate the prevalence of incontinence in individuals aged 11–20 years to be around 2%–3% for daytime wetting, 2%–2.5% for bedwetting and 1%–1.5% for soiling.[5 6] Incontinence in children and young people has been linked to high rates of emotional distress, poor self-image, relationship problems and unhappiness at school.[7–12] Poor management of continence problems has an adverse impact on quality of life and self-esteem during childhood and adolescence.[13]

A minority of children and young people have an organic (neurological, structural or anatomic) cause for their incontinence, but

the vast majority of paediatric continence problems are functional (no underlying organic cause).[1 14 15] Uncertainty surrounding individual responses to treatments for functional incontinence and varying levels of access to paediatric continence services are major factors affecting patients with incontinence.[2] Treatment for functional incontinence is varied, and there is often no singularly effective treatment. As a consequence, patients may need to try a series of different approaches to manage or improve their incontinence symptoms.[16 17] The first-line treatment for daytime wetting is often urotherapy, involving a structured regime of toileting and fluid intake, sometimes combined with medication.[18–20] Initial treatments for bedwetting include the bedwetting alarm and/or medication (eg, desmopressin).[14 16 21] Constipation is a common cause of soiling and can exacerbate urinary incontinence due to pressure on the bladder. The first stage of treatment for soiling usually involves laxatives or stool softeners in addition to a toileting programme and, in some cases, suppositories.[15 22 23]

Although many children experience a natural resolution of their incontinence with age, there is evidence that those with severe incontinence are at risk of their problems persisting into adolescence.[6 24–27] Often when curative treatment fails there is a refocusing on symptom management as the child continues to mature and enters into adolescence. Adolescence can be a particularly challenging time for the management of chronic health problems and is frequently associated with lower levels of treatment adherence.[28] Due to the challenges of treatment, young people with persistent continence problems may see minimal improvements and may need to follow treatment regimes for extended periods before seeing any changes. Previous work has shown that adolescents often struggle to adhere to treatments that show few immediate benefits.[28 29]

Little research has been published on the primary and secondary care experiences of young people with continence problems, especially with regard to their experiences of treatment and management of their continence problems. The overall aim of the project was to explore the impact of continence problems on young people, and the specific focus of this paper is on their experiences of primary and secondary care.

## METHODS
Participants were recruited through five secondary care paediatric continence clinics (four in England and one in Scotland) and through an advertisement on a paediatric continence charity website (ERIC; The Children's Bowel and Bladder Charity, www.eric.org.uk). Eligible participants were between 11 and 20 years, currently experiencing continence problems (bedwetting, daytime wetting or soiling) or who previously experienced these problems after the age of 10 years and and able to speak and understand English.

Participants attending paediatric continence clinics were given a study information pack by their clinician. Those recruited through the advertisement were sent an information pack by the research team. Ethical approval was given by the National Research Ethics Service Committee South West – Central Bristol (14/SW/0059). The researcher contacted all interested participants by phone to describe the study, the motivation for the work and answer any questions. All participants were unknown to the researcher prior to the commencement of the study. In total, 45 potential participants were identified (36 through clinics and 9 through ERIC). Twenty-five participants were not interviewed for the following reasons: 12 unable to contact, 5 not interested in taking part and 8 withdrawn.

In-depth semistructured interviews were conducted with 20 young people between February 2015 and January 2016 (see table 1 for participant characteristics). Recruitment and final sample size were guided by the concept of 'information power'.[30] Given the specificity of the sample, broad study aim and strong interview dialogue, a sample size of 20 participants was considered appropriate.

Participants were given the option to be interviewed by telephone, Skype or face to face (for participants within a 40-mile radius of Bristol). Among the participants, 11 were interviewed by Skype and 9 by telephone. No participants were interviewed face to face (those given this option preferred to be interviewed by Skype or telephone). No repeat interviews were conducted. Four participants chose to have their parent/guardian present during the interview, seven were in a communal area of their house and nine chose to conduct the interview alone in a private room.

A topic guide was developed based on previous literature on the treatment, management and impact of continence problems in children and young people. The topic guide covered issues such as attending appointments, treatment experiences, school or work and thoughts and feelings about their continence problem. Due to the exploratory nature of the study, the interviews were semistructured. The topic guide was used as a starting point for discussion, with flexibility to discuss novel areas introduced by the participants.

An arts-based participatory approach was used in the interviews. This approach is considered appropriate for children and young people since it provides additional narratives through which personal experiences can be explored.[31] A participant activity pack was developed for use prior to and during the interviews containing a graphic representation of each possible topic area, allowing the participants to write or draw their thoughts in advance, if they wished to. Participants were sent the pack in advance of their interview and were given a verbal explanation of how it could be used during the initial phone call.

All participants provided written informed consent, for ages 16 and above, or parental written consent and child assent for those below 16 years. Interviews were

**Table 1** Participant characteristics

| Participant ID no. | Gender | Age group* | Continence problem | Method of interview |
|---|---|---|---|---|
| 1 | M | 11–13 | Night wetting | Skype |
| 2 | M | 14–16 | Day and night wetting | Telephone |
| 3 | F | 17–19 | Daytime wetting | Skype |
| 4 | F | 14–16 | Daytime wetting | Skype |
| 5 | F | 11–13 | Day and night wetting | Skype |
| 6 | M | 14–16 | Day and night wetting | Skype |
| 7 | M | 11–13 | Night wetting | Telephone |
| 8 | M | 11–13 | Night wetting | Telephone |
| 9 | M | 11–13 | Day and night wetting | Telephone |
| 10 | F | 14–16 | Night wetting | Skype |
| 11 | M | 14–16 | Daytime wetting | Skype |
| 12 | M | 17–19 | Night wetting | Telephone |
| 13 | F | 11–13 | Soiling | Skype |
| 14 | F | 11–13 | Soiling | Skype |
| 15 | F | 14–16 | Daytime wetting | Skype |
| 16 | F | 14–16 | Daytime wetting | Telephone |
| 17 | M | 11–13 | Soiling | Telephone |
| 18 | F | 17–19 | Night wetting | Telephone |
| 19 | M | 11–13 | Soiling | Telephone |
| 20 | M | 14–16 | Day and night wetting | Skype |

*For confidentiality reasons, participant age is given in group.
M, male; F, female.

conducted by one female researcher with extensive experience of qualitative work (KW: DHealthPsy, Senior Research Associate) and lasted between 34 and 99 min (mean 65 min). The researcher made detailed field notes after each interview, including details about the environment, participant demeanour and personal reflections. Data were recorded using an encrypted audio recorder. Data collection and analysis were conducted in parallel after completion of the first five interviews. Early analysis was used to refine the topic guide and to further explore emerging areas of interest. Interviews were audio recorded, fully transcribed and imported into the software package NVivo10. Inductive thematic analysis was carried out following guidelines of Braun and Clarke.[32] Following completion of the first five interviews, each transcript was read, and the data were free coded across all transcripts. A selection of three transcripts were also independently free coded by the study team (CJ and HC). Codes were discussed and compared with all members of the team in order to further refine coding and to maximise rigour.[33] An initial set of agreed codes were set up within the NVivo10 database, and any new codes identified from further interviews were discussed within the team and added to the coding framework. A strong theme was identified from the data on young people's experiences of clinical care.

### Sample description
In total, 20 interviews were carried out with children and young people aged 11–20 years. Seventeen were recruited through paediatric continence clinics and three through the ERIC advertisement. Three had organic incontinence and 17 had non-organic incontinence. Table 1 provides a full overview of participant characteristics.

### RESULTS
#### Themes
Three themes directly linking to clinical care experiences were identified from the data: *appointment experiences*, *treatment experiences* and *engagement with treatment*. A fourth theme was identified from data that was not directly linked to clinical experiences but was felt to have important implications for self-management and adherence: *internalisation and externalisation of the continence problem*.

#### Appointment experiences: what makes a good appointment?
Participants in the study had seen a range of health professionals during their treatment journey including general practitioners (GPs), community and school nurses, paediatricians, urologists and paediatric continence specialists. Participants' first contact with health services was most commonly with a GP or a community or school nurse. Participants recruited through the paediatric continence

clinics had the experience of attending a specialist service, while those recruited through the online advertisement had primarily seen their GP or a small number had seen a general paediatrician.

For all participants, the most important factor that influenced their appointment experiences was the communication style used by the clinician, specifically, whether the clinician used patient-centred communication and age-appropriate language. Participants expressed a strong desire for the clinician to include them in the conversation and talk to them directly (rather than to their parent) and to use language that they could understand.

"The language they used, and everything, made it easier. It made it kind of child friendly I suppose, which is what they should be doing".—P12, male, aged 19 years, night wetting

When health professionals failed to do this, participants talked about feeling confused and annoyed.

"I always had to ask my mum 'what does that mean?' […] he didn't even explain what it was and even my mum didn't really know what it was half the time, because they say things that they think that you'll know, but you don't know. That's frustrating because it's like, I don't know what that means and they're not telling me what that means". —P14, female, aged 12 years, soiling

Due to the sensitive nature of continence problems, appointments often involved questions and discussions about very personal and potentially embarrassing issues. Therefore, building trust and rapport with the clinician was especially important for young people. Continuity of care was highlighted as a factor that helped build rapport and was highly desirable. Participants who reported seeing the same clinician at each appointment described how this helped them to feel more comfortable.

"After my second visit I got to know her a little better and I felt more comfortable every time I went".—P9, male, aged 11 years, day and night wetting

Continuity of care was more likely when participants had been referred to a specialist continence clinic but was not always guaranteed. More than half of the participants described seeing a different clinician on each visit with no explanation as to why. Participants said that new clinicians were not familiar with their case history or gave conflicting advice to previous clinicians.

"when we're at the hospital it's always a new person and they always say a different thing to what the other doctor said".—P1, male, aged 12 years, night wetting

Appointments often centred on developing treatment plans to be implemented independently at home by the young person and/or parent. Participants expressed a strong desire to be included in the decision-making process, through being given a rationale for each treatment option and a degree of choice in how to use the treatment plan at home. When health professionals engaged in shared decision-making approaches, participants reported positive appointment experiences.

"I like to know what to do and why I should do it".— P8, male, aged 11 years, night wetting

"He's just really warm and stuff and he just says 'How are you feeling? Do you want to do that? Do you not want to do that?'".—P14, female, aged 12 years, soiling

Participants described wanting to be treated as an intelligent person and not wanting to be patronised by their clinician. Being seen as an expert in relation to themselves and their body was important.

"I like them to treat me as if I'm the person in question, because I'm not stupid, I know I'm not brilliant at medicine or anything, but I can do my research. I know about the medics of it. I know about the treatments that I had, and I know about me, more than anyone".—P2, male, aged 14 years, day and night wetting

### Treatment experiences: the frustration of no magic bullets

Participants often recalled trying a wide range of treatments including urotherapy, dietary changes, medication(s), bedwetting alarms, Botox injections into the bladder, laxatives, suppositories, TENS (transcutaneous electrical nerve stimulation) machine and hypnotherapy. Almost all participants had tried more than one treatment and had experienced some form of relapse or complete treatment failure. Participants explained that with each failed treatment or relapse, their belief in the success of subsequent treatments diminished.

"For the first four months, it was fine…it was a lot better. Then it started to get worse again, and by six or seven months, it was actually worse than it was before I had the [Botox] operation. We went back to do a second one…it was fine for six months. Then it got worse again, and then they called us back to have another one, and we said no. By that point, I was kind of sick of it, and I knew that it wasn't going to work".—P2, male, aged 14 years, day and night wetting

Belief in treatments seemed to be made much more difficult because of the uncertainty of the cause of young people's continence problems. A minority of participants had been given a medical explanation for their incontinence and reported that this was helpful with respect to the treatments then offered.

"I think it made me more comfortable, knowing it was actually a problem and it wasn't just my mum being frantic or something".—P4, female, aged 14 years, daytime wetting

The majority of participants, however, were given no medical explanation for their incontinence. Participants said that they found this frustrating and confusing.

"I really struggled, just thinking, 'Why has it become a problem for me?' Because there was not any reason".—P3, female, aged 18 years, daytime wetting

Not being able to match suggested treatments to an underlying cause of the incontinence was also challenging for participants to understand. Other participants had been told there was nothing wrong with them.

"Him saying there's nothing wrong with me, he made me feel quite vulnerable […] it really made me feel bad because we were told that he is the man to see… if he can't do anything about it, like who can?".—P11, male, aged 14 years, daytime wetting

Participants talked extensively about their emotional reactions to treatments saying that the lack of explanation, unpredictability of treatment success combined with experiences of relapse and treatment failure led them to feel upset and cynical about the chances of resolution. Many described giving up hope that their problem would ever go away.

"Some of it's obviously a little bit upsetting because I've had so many different medications and trying new things and it hasn't really worked yet, so my hope of it going to getting smaller and smaller".—P16, female, aged 15 years, daytime wetting

For older participants, who had often been undergoing treatment for several years, feelings of pessimism were particularly common.

"For quite a lot of the time the, I felt really really pessimistic about this problem, just like there is absolutely no hope that it's ever going to be better and I'm going to be stuck with this problem forever. Clearly I don't want that to happen, so that really upsets me about it".—P3, female, aged 18 years, daytime wetting

### Engagement with treatment: the effects of disappointing results

Participants reported varying levels of engagement with their treatment. Experiencing relapse or treatment failure led many participants to disengage from their treatment, with some admitting that they avoided taking their medication or lied about this to their parents.

"Instead of taking the tablets, I think mum thought I was doing it, say, once a week or something, I was [throwing them away] every day and having problems every day because of that".—P4, female, aged 14 years, daytime wetting

One participant talked about feeling so frustrated with the lack of improvement in her symptoms that she decided no treatment was going to work and there was no point trying anything new.

"I think it was because I'd been on them [the medication] for so long, I didn't feel like they were doing anything. I've never actually tried it [the new medication]. I've never actually properly tried it […] I just decided that it wouldn't work either".—P4, female, aged 14 years, daytime wetting

Another participant described disengaging from treatment not due to lack of belief in its efficacy but as an effort to regain her independence.

"I think there was a period when I was in year nine [aged 13–14] where I didn't take my medication very well and it just got worse. […] I was just like 'I can do it on my own, I'm an independent woman, I can do what I want'".—P15, female, aged 14 years, daytime wetting

### Internalisation and externalisation of the continence problem

Through discussion with participants about their beliefs and attitudes towards their continence problem, two contrasting coping mechanisms were identified: internalisation and externalisation of the continence problem. Participants who internalised their continence problem integrated it into their self-identity. They viewed the problem as a part of their lives and themselves that needed to be actively managed but not as something that defined or controlled them. Participants who demonstrated internalisation talked about a process of accepting that they had a continence problem and the need to actively manage it. Taking ownership of their continence problem and embracing it because it made them 'different' and 'unique' were common lines of thought. Others talked about how this attitude helped them to cope with worries about other people finding out about their problem.

"I kind of realised that, actually, it's something I can't change, and it's a part of me, so I shouldn't be scared of people knowing. It's not something I've done wrong".—P2, male, aged 14 years, daytime and night wetting

In contrast, those who externalised their continence problem saw it as something separate to themselves and their identity and as an intruder into their lives. Participants who demonstrated externalisation often avoided thinking about their continence problem and talked about distancing themselves from their problem and rejecting it from their self-identity.

"…it's just frustrating also because you know what's going to happen. So you kind of always, that's always at the back of your mind. So I do then just try and avoid it [thinking about it] if I can".—P12, male, aged 19 years, night wetting

In order to deal with stressful social situations, such as sleepovers or school trips, participants said they tried

to convince themselves it would just be fine, even if they knew this was highly unlikely.

> "I think when you have this issue you're trying to put aside, you know the problems but you're trying to get it out of your head, so you will do all you can to convince yourself that you'll be fine even if you won't be".—P11, male, aged 14 years, night wetting

## DISCUSSION
### Appointment experiences
Through interviews with young people with continence problems, we find that positive treatment experiences are characterised by seeing the same clinician at each appointment and by the use of age-appropriate communication. Building a relationship with the same clinician was desirable as it facilitated greater trust and rapport and supported young people to feel more comfortable disclosing highly personal information. Participants expressed a strong desire to be given a full explanation of their continence problem, appropriate to their age and level of understanding. This finding is consistent with studies of clinical experiences of young people with other chronic health problems.[34–37] Young people in this study wanted to be involved in making decisions about their treatment options, rather than the traditional focus on parent–doctor communication in paediatric consultations. Aligned to this, it is now recommended that children and young people should be the primary focus of their own appointments.[16 38] There is evidence that this facilitates better communication and patient understanding, supports patient autonomy and competence and promotes successful behaviour change and illness management.[39 40] Shared decision making between clinician and patient could aid treatment engagement and autonomy in young people and provide them with skills and confidence to transition into adult care.

### Factors affecting treatment engagement
The majority of participants in this study had functional incontinence (n=17). In the absence of an underlying organic cause of the incontinence, clinicians were often unable to give specific guidance on treatments and prognosis, creating significant illness uncertainty for the participants.[41 42] This presents challenges since participants often had to try a range of treatments, with varying success.[1] Treatment failures and relapses were common and undermined participants' beliefs in future treatment success and the controllability of their continence problem.

Illness uncertainty and lack of belief in the controllability of a health problem are significant barriers to adherence and engagement, because individuals are less likely to follow a treatment regime they do not believe will work.[43 44] Models of health behaviour change consistently highlight the perceived controllability of a health problem as a key factor in predicting health behaviours

and treatment adherence.[45 46] Additionally, illness uncertainty has been linked to maladaptive coping, increased psychological distress, depression and reduced quality of life.[47] When illness uncertainty is an issue, it is crucial that clinicians work to positively engage patients and support them in long-term self-management in order to minimise the impact of their health problem on daily life. Successful self-management also promotes great individual autonomy and can help patients to feel more in control of their bodies.[48] Young people with functional incontinence could also benefit from psychological support and interventions to manage uncertainty and reduce anxiety, such as acceptance commitment therapy.[49] Finding a balance between giving patients hope and being realistic about treatment success is vital in managing patient expectations and could help to mitigate against negative treatment experiences. By increasing understanding of how treatment failures and relapses affect patients, health professionals can identify potential periods of high-risk non-adherence and take steps to manage this.

### Internalisation and externalisation of the problem
Among the participants in our sample, there was a dichotomy between those who demonstrated internalisation and externalisation of their continence problem. Those who internalised their continence problem had more successfully incorporated this into their self-identity. They recognised it as an ongoing part of their lives but did not feel restrained or defined by it. In contrast, those who externalised their problem saw it as an external factor that intruded into their life, and avoidance coping was common in this group.

Although participants did not directly link these coping strategies to their clinical experiences, the way in which they view their health condition does have significant implications for self-management and adherence to treatment. By viewing continence problems as external to the self, this undermines beliefs in the controllability of the condition, which are linked to treatment adherence.[45 46 50] If an individual engages in avoidance coping, they are likely to avoid any stimuli that reminds them of their continence problem, including treatment regimes. Non-adherence increases the risk of treatment failure or relapses, which may further reinforce the belief that the condition is uncontrollable, creating a self-perpetuating cycle of disengagement and treatment failure. In contrast, acceptance of a chronic health problem and integration of illness experiences where the condition is viewed as only one characteristic of an individual's life are linked to a greater sense of personal control and proactive management behaviours.[51–53]

While within our sample internalisation and externalisation are seen as two different groups, it is likely that within the wider population these responses fall along a continuum. Additionally, individuals may be likely to switch between the two attitudes depending on their experiences. Experiencing relapse or feeling overwhelmed by the condition may promote more externalising attitudes

in order to cope. Alternatively, successful treatment and management symptoms may support greater internalisation. Without more exploration of these issues, the exact relationships and mechanism at work are not clear, and further work is needed to better understand this.

## Limitations

Only a small number of participants with soiling were recruited into this study. Young people with soiling problems reported some different experiences to those with urinary incontinence, suggesting that further exploration is needed. Clinicians involved in recruiting to the study reported that young people with soiling were generally less engaged during appointments and would often not admit to having a problem. This led to a problem recruiting these individuals. The participants with soiling either had a medical explanation for their problem (n=1) or were experiencing significant improvements in their symptoms (n=3). Another limitation of the sample was that all the participants in this study were receiving medical help for their continence problem and the majority were seeing specialists. Young people who have not received specialist help for their continence problems are likely to have different experiences to the participants in this study. This study could not explore cultural differences in young people's experiences of continence problems since all but one of the participants identified as white British. In addition, as all participants were from the UK, we do not know if the same issues are experiences in different countries. Socioeconomic background could also impact on experiences of young people with continence problems, but these data were not collected in the current study. As interviews were conducted by Skype and telephone, it is possible that some nuances of the participants' responses and body language may have been missed. Although this does not change the results, face-to-face interviews may have yielded an additional level of interpretation and insight.

## CONCLUSIONS AND FUTURE DIRECTIONS

Incontinence is challenging to treat and manage, particularly when there is no organic cause. Long-term self-management and helping individuals regain a sense of control is key in minimising the impact of incontinence on daily life. Ensuring positive appointment experiences and engaging young people in their own healthcare is vital. A focus on shared decision making and ensuring continuity of care are both highly important and facilitate greater engagement with treatment. Clinicians should use age-appropriate language to ensure patient understanding and work to build rapport in order to promote trust and disclosure. Relapses or treatment failure are common and can have a negative emotional impact and undermine beliefs in future treatment success. Assessment of coping styles may a beneficial addition to clinical assessment. For patients who are disengaged or have low treatment

adherence, providing psychological support to promote acceptance may be an important first step in successful long-term self-management. These results demonstrate that the same challenges are experienced across the age range and for both organic and non-organic continence problems; this reinforces the importance of addressing these issues.

**Contributors** KW: lead researcher, design of qualitative work, conducting interviews, analysis of interview data, writing and preparing manuscript. HC: design of qualitative work, analysis of a subset of interviews and input into analysis process, member of the project steering group, providing comments and feedback on manuscript. AW: recruitment of participants, contribution to interview design, contribution to interpretation of themes, member of the project steering group and revising manuscript critically for important intellectual content. CS: contribution to interpretation of themes and revising manuscript critically for important intellectual content. CJ: principal investigator, obtaining ethical approval for the study, design of qualitative work, analysis of a subset of interviews and input into analysis process, providing comments and feedback on manuscript.

**Funding** This research was funded by a grant from the Medical Research Council (Increasing understanding of risk factors and outcomes associated with continence problems in children and adolescents. MRC reference: MR/L007231/1).

**Competing interests** None declared.

**Ethics approval** National Research Ethics Service Committee South West - Central Bristol.

**Provenance and peer review** Not commissioned; externally peer reviewed.

**Data sharing statement** Data from the study are classified as controlled access. No participant consent was given to share their interview data; therefore, no further data from this study are available.

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
