## [Reviewer comments · BMJ Open]

ARTICLE DETAILS

TITLE (PROVISIONAL)	'What does that mean?': A qualitative exploration of the primary and secondary clinical care experiences of young people with continence problems in the UK
AUTHORS	Whale, Katie; Cramer, Helen; Wright, Anne; Sanders, Caroline; Joinson, Carol

VERSION 1 - REVIEW

REVIEWER	Janet Chase Victorian Children's Continence Clinic, Cabrini Hospital, Melbourne, Victoria, Australia. Paediatric Gastroenterology Victoria, Royal Children's Hospital, Melbourne, Victoria, Australia.
REVIEW RETURNED	06-Feb-2017

GENERAL COMMENTS	A few corrections Abstract line 24 ? should read 'patient focussed appointments page 5 line 45. I do not agree that first line treatment is laxatives alone -it is laxatives and a toileting programme Page 17 line 8 ? should read 'greater sense of personal control' line 12 ? should read 'are seen' line 17 ? should read 'to a problem' For a long time the message from ICCS has been -"talk to the child, engage the child, age appropriate education regarding the problem and treatment, and in so doing remove blame and guilt" This paper is a timely and humbling reminder that even in specialist clinics we fall down on the very things that should be at the centre of our interventions.
--

REVIEWER	Stuart B. Bauer Boston Children's Hospital Boston, MA USA
REVIEW RETURNED	06-Feb-2017

GENERAL COMMENTS	The authors of this manuscript have done an outstanding job in looking at an important issue regarding how older children and adolescents view their incontinence and their relationship with their providers. They have done a commendable job in organizing the project and conducting interviews with 20 participants. Their manuscript is clear and although long, concise in its presentation, as they have a complicated message to convey. I feel this is one of the few manuscripts I have had to review over the years in which the authors addressed all potential ambiguities in their submitted version. The only comment I would make is regarding their
---

	limitations section. As most of their interviews were conducted via SKYPE and not face-to-face, they may have missed some nuances in respondents' answers that might be more evident had the interviews been truly live. Would the authors consider making a comment regarding this issue in their 'limitations' section? Also, the authors need to mention they don't know what cultural differences there may be if the same study was conducted in another country. These are very moot points that shouldn't detract from the Journal accepting this manuscript for publication.
--	--

REVIEWER	Francois Cachat Department of Pediatrics Division of pediatric nephrology University Hospital Lausanne Switzerland
REVIEW RETURNED	17-Apr-2017

GENERAL COMMENTS	What does that mean? Clinical care experience of young people with continence problems. In their original qualitative research, Katie Whale et al report their findings after interviewing young adolescents with enuresis / encopresis problems. This is a well performed study, instructive, probably very useful for young (and no so young) doctors looking after such patients to help them delivering a patient focused health care. After carefully reading this paper, I only have a few minor comments: Title: Appropriate. If feasible, it would be useful for readers / reviewer to find the word "qualitative" in the title. This is only a suggestion. Abstract: Appropriate. In the settings, line 14 I would mention: primary and secondary care in the UK. Findings might differ in different countries. Introduction: Appropriate. Reference 2 refers to a number of children with incontinence in the UK, however reference 2 is probably not (looking at the reference table) the original paper citing this number. When feasible, especially with references dealing with "hard data" or numbers, I like to cite the original paper. If that is the original paper, that's fine. In the same vein, the message with reference 28 could be reinforced with a reference actually mentioning percentage of adherence in children with enuresis : Dieter Baeyens, Anneleen Lierman, Herbert Roeyers, Piet Hoebeke, Johan Vande Walle. Adherence in children with nocturnal enuresis. J Pediatr Urol 2008). Methods: Good. It would be interesting to know how many children were approached for the study, how many refused, retracted. Obviously not possible for the "advert-reached" children /families. Also, comparing demographic between participants and non participants
---

	would be interesting, but I understand these data were not collected. Table 1: Good. I question myself, are patients 18 and 19 years old really similar to the young ones? Probably not, in that sense they probably “carry” years of relapse behind them, compared to younger ones /11-12 years old). However it is fascinating to see they report the same complaints. This might be mentioned in the text? It reinforces the message. In the same vein, patients with organic cause are probably slightly different, but (see above) they convey the same message. Results: Appropriate. Discussion/limitation: Appropriate. Conclusion: If not too long, and knowing that 50% of readers just read the conclusion..... I would try to include as well, if feasible, the aspects of continuity of care, of shared decision and of clear explanation that were evoked in the answers of the participants. References: See above in introduction. Also, ref 24 to 27 have “numbers” that have nothing to do there.
--	---

VERSION 1 – AUTHOR RESPONSE

Reviewer 1

1. Abstract line 24 ? should read 'patient focussed appointments

Response: This has been added

2. page 5 line 45. I do not agree that first line treatment is laxatives alone -it is laxatives and a toileting programme

Response: 'in addition to a toileting programme' has been added

3. Page 17 line 8 ? should read 'greater sense of personal control'

Response: This has been corrected

4. line 12 ? should read 'are seen'

Response: This has been corrected

5. line 17 ? should read 'to a problem'

Response: This has been corrected

Reviewer 2

1. I would make is regarding their limitations section. As most of their interviews were conducted via SKYPE and not face-to-face, they may have missed some nuances in respondents' answers that might be more evident had the interviews been truly live. Would the authors consider making a comment regarding this issue in their 'limitations' section?

Response: This is a very good point and something which we discussed in detail within the team.

Conducting the interviews by Skype and telephone enabled us to increase our recruitment areas and final numbers, however we agree that face to face interviews may have given us more insight into the participants' responses and emotional reactions. To acknowledge this the following text has been added to the limitations section: "As interviews were conducted by Skype and telephone it is possible that some nuances of the participants' responses and body language may have been missed. Although this does not change the results, face to face interviews may have yielded an additional level of interpretation and insight."

2. The authors need to mention they don't know what cultural differences there may be if the same study was conducted in another country.

Response: We have further clarified this point but adding the following text: "In addition, as all participants were from the UK, we do not know if the same issues are experienced in different countries.". This is an interesting point and we would be keen to explore cross-cultural differences in future work.

Reviewer 3

Title:

1. Appropriate. If feasible, it would be useful for readers / reviewer to find the word "qualitative" in the title. This is only a suggestion.

Response: The word 'qualitative' has been added to the title

Abstract:

2. Appropriate. In the settings, line 14 I would mention: primary and secondary care in the UK. Findings might differ in different countries.

Response: This has been added

Introduction:

3. Appropriate. Reference 2 refers to a number of children with incontinence in the UK, however reference 2 is probably not (looking at the reference table) the original paper citing this number. When feasible, especially with references dealing with "hard data" or numbers, I like to cite the original paper. If that is the original paper, that's fine.

Response: The reference has been changed to the original data source: Public Health England. Continence Needs Assessment module. 2013. Available: <http://atlas.chimat.org.uk/IAS/profiles/needsassessments>

4. In the same vein, the message with reference 28 could be reinforced with a reference actually mentioning percentage of adherence in children with enuresis : Dieter Baeyens, Anneleen Lierman, Herbert Roeyers, Piet Hoebeke, Johan Vande Walle. Adherence in children with nocturnal enuresis. J Pediatr Urol 2008).

Response: This reference has been added and numbers updated accordingly

Methods:

5. Good. It would be interesting to know how many children were approached for the study, how many refused, retracted. Obviously not possible for the "advert-reached" children /families. Also, comparing demographic between participants and non participants would be interesting, but I understand these data were not collected.

Response: The following text has been added with the requested information: "In total 45 potential participants were identified (36 through clinics and 9 through ERIC). 25 participants were interviewed

for the following reasons: 12 unable to contact, 5 not interested in taking part, 8 withdrawn.”

Table 1:

6. Good. I question myself, are patients 18 and 19 years old really similar to the young ones? Probably not, in that sense they probably “carry” years of relapse behind them, compared to younger ones /11-12 years old). However it is fascinating to see they report the same complaints. This might be mentioned in the text? It reinforces the message.

In the same vein, patients with organic cause are probably slightly different, but (see above) they convey the same message.

Response: We agree. The participants in the study all have different journeys and experiences with their continence problems, older participants are likely to have experiences multiple relapses over the years. However, the results show that the same challenges are reported across the age range and for organic and non-organic incontinence. We have added the following sentence to the conclusion to reflect this: “These results demonstrate that the same challenges are experienced across the age range and for both organic and non-organic continence problems; this reinforces the importance of addressing these issues.”

Conclusion:

7. If not too long, and knowing that 50% of readers just read the conclusion..... I would try to include as well, if feasible, the aspects of continuity of care, of shared decision and of clear explanation that were evoked in the answers of the participants.

Response: To include these points the following text has been added to the conclusion: “Ensuring positive appointment experiences and engaging young people in their own healthcare is vital. A focus on shared decision-making and ensuring continuity of care are both highly important and facilitate greater engagement with treatment. Clinicians should use age-appropriate language to ensure patient understanding and work to build rapport in order to promote trust and disclosure.”

References:

8. See above in introduction. Also, ref 24 to 27 have “numbers” that have nothing to do there.

Response: The numbers have been removed and other reference comments as addressed as previously stated.

VERSION 2 – REVIEW

REVIEWER	Stuart B. Bauer Boston Children's Hospital Department of Urology Boston, MA 02115 USA
REVIEW RETURNED	19-May-2017

GENERAL COMMENTS	The authors have addressed the minor concerns I had when I read the original submission of the manuscript. They have adequately answered all my concerns in this revised paper.
---

REVIEWER	Francois Cachat University Hospital Department of Pediatrics 1011 Lausanne Switzerland
-----------------	--

REVIEW RETURNED	01-May-2017
-------------

GENERAL COMMENTS	page 7 line 27 in yellow, should read:25 participants were NOT interviewed for the following reasons.....
--

VERSION 2 – AUTHOR RESPONSE

Reviewer 3 comment: page 7 line 27 should read '25 participants were NOT interviewed..
This has been corrected.